# Once Quantized for All: Progressively Searching for Quantized Compact Models

## Abstract

Automatic search of Quantized Neural Networks (QNN) has attracted a lot of attention. However, the existing quantization-aware Neural Architecture Search (NAS) approaches inherit a two-stage search-retrain schema, which is not only time-consuming but also adversely affected by the unreliable ranking of architectures during the search. To avoid the undesirable effect of the search-retrain schema, we present Once Quantized for All (OQA), a novel framework that searches for quantized compact models and deploys their quantized weights at the same time without additional post-process. While supporting a huge architecture search space, our OQA can produce a series of quantized compact models under ultra-low bit-widths(*e.g.* 4/3/2 bit). A progressive bit inheritance procedure is introduced to support ultra-low bit-width. Our searched model family, OQANets, achieves a new state-of-the-art (SOTA) on quantized compact models compared with various quantization methods and bit-widths. In particular, OQA2bit-L achieves 64.0% ImageNet Top-1 accuracy, outperforming its 2 bit counterpart EfficientNet-B0@QKD by a large margin of 14% using 30% less computation cost.

## 1 Introduction

Compact architecture design (Sandler et al., 2018; Ma et al., 2018) and network quantization methods (Choi et al., 2018; Kim et al., 2019; Esser et al., 2019) are two promising research directions to deploy deep neural networks on mobile devices. Network quantization aims at reducing the number of bits for representing network parameters and features. On the other hand, Neural Architecture Search(NAS) (Howard et al., 2019; Cai et al., 2019; Yu et al., 2020) is proposed to automatically search for compact architectures, which avoids expert efforts and design trials. In this work, we explore the ability of NAS in finding quantized compact models and thus enjoy merits from two sides. Traditional combination of NAS and quantization methods could either be classified to NAS-then-Quantize or Quantization-aware NAS as shown in Figure 1.

Conventional quantization methods merely compress the off-the-shelf networks, regardless of whether it is searched (EfficientNet (Tan & Le, 2019)) or handcrafted (MobileNetV2 (Sandler et al., 2018)). These methods correspond to NAS-then-Quantize approach as shown in Figure 1(a). However, it is not optimal because the accuracy rank among the searched floating-point models would change after they are quantized. Thus, this traditional routine may fail to get a good quantized model. Directly search with quantized models' performance seems to be a solution.

Existing quantization-aware NAS methods (Wang et al., 2019; Shen et al., 2019; Bulat et al., 2020; Guo et al., 2019; Wang et al., 2020) utilize a two-stage search-retrain schema as shown in Figure 1(b). Specifically, they first search for one architecture under one bit-width setting[1], and then retrain the model under the given bit-width. This two-stage procedure undesirably increases the search and retrain cost if we have multiple deployment constraints and hardware bit-widths. Furthermore, due to the instability brought by quantization-aware training, simply combining quantization and NAS results in unreliable ranking (Li et al., 2019a; Guo et al., 2019) and sub-optimal

---

[1]One bit-width setting refers to a specific bit-width for each layer, where different layers could have different bit-widths.

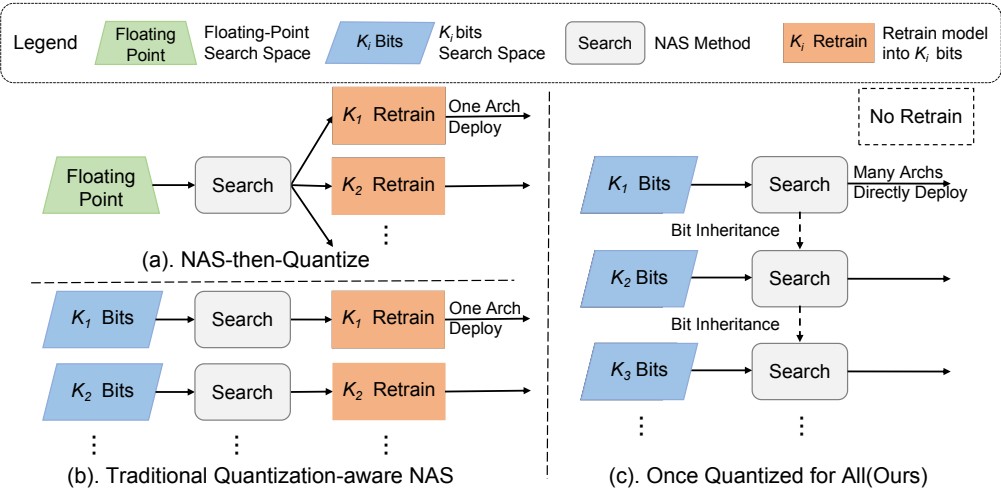

Figure 1: The overall frameworks of existing works on combining quantization and NAS and our method. (a) denotes directly converting the best searched floating-point architecture to quantization. (b) first adopts a quantization-aware search algorithm to find a single architecture, then retrain the quantizaed weights and activation. Our OQA (c) can search for many quantized compact models under various bit-widths and deploy their quantized weights directly.

quantized models (Bulat et al., 2020). Moreover, when the quantization bit-width is lower than 3, the traditional training process is highly unstable and introduces very large accuracy degradation.

To alleviate the aforementioned problems, we present Once Quantized for All (OQA), a novel framework that: 1) searches for quantized network architectures and deploys their quantized weights immediately without retraining, 2) progressively produces a series of quantized models under ultra-low bits(*e.g.* 4/3/2 bit). Our approach leverages the recent NAS approaches which do not require retraining (Yu & Huang, 2019; Cai et al., 2019; Yu et al., 2020). We adopt the search for kernel size, depth, width, and resolution in our search space. To provide a better initialization and transfer the knowledge of higher bit-width QNN to the lower bit-width QNN, we propose bit inheritance mechanism, which reduces the bit-width progressively to enable searching for QNN under different quantization bit-widths. Benefiting from the no retraining property and large search space under different bit-widths, we can evaluate the effect of network factors.

Extensive experiments show the effectiveness of our approach. Our searched quantized model family, OQANets, achieves state-of-the-art (SOTA) results on the ImageNet dataset under 4/3/2 bit-widths. In particular, our OQA2bit-L far exceeds the accuracy of 2 bit Efficient-B0@QKD (Kim et al., 2019) by a large 14% margin using 30% less computation budget. Compared with the quantization-aware NAS method APQ (Wang et al., 2020), our OQA4bit-L-MBV2 uses 43.7% less computation cost while maintaining the same accuracy as APQ-B.

To summarize, the contributions of our paper are three-fold:

- Our OQA is the first quantization-aware NAS framework to search for the architecture of quantized compact models and deploy their quantized weights without retraining.

- We present the bit inheritance mechanism to reduce the bit-width progressively so that the higher bit-width models can guide the search and training of lower bit-width models.

- We provide some insights into quantization-friendly architecture design. Our systematical analysis reveals that shallow-fat models are more likely to be quantization-friendly than deep-slim models under low bit-widths.

## 2  RELATED WORK

**Network Architecture Search without retraining**  Slimmable neural networks (Yu et al., 2018) first propose to train a model to support multiple width multipliers(*e.g.* 4 different global width

multipliers for MobileNetV2). OFA (Cai et al., 2019) and BigNAS (Yu et al., 2020) push the envelope forward in network architecture search (NAS) by introducing diverse architecture space (stage depth, channel width, kernel size, and input resolution). These methods propose to train a single over-parameterized supernet from which we can directly sample or slice different candidate architectures for instant inference and deployment. However, all of the aforementioned methods are tailored towards searching for floating-point compact models. Converting the best floating-point architecture to quantization tends to result in sub-optimum quantized models. In the quantization area, recent papers (Jin et al., 2019a; Yu et al., 2019) propose to train a single model that can support different bit-widths. But they only quantize manually design networks(*e.g.* ResNet, MobileNetV2) under relatively high bit-widths (*e.g.* 4 bit for MobileNetV2), our OQA can search for architectures and produce quantized compact models with lower bit-width(*e.g.* 2 bit).

**Quantization-aware Network Architecture Search**   Recent studies combine network quantization and NAS to automatically search for layer bit-width with given architecture or search for operations with given bit-width. HAQ (Wang et al., 2019) focuses on searching for different numbers of bits for different layers in a given network structure and shows that some layers, which can be quantized to low bits, are more robust for quantization than others. AutoBNN (Shen et al., 2019) utilizes the genetic algorithm to search for network channels and BMobi (Phan et al., 2020) searches for the group number of different convolution layer under a certain 1 bit. SPOS (Guo et al., 2019) trains a quantized one-shot supernet to search for bit-width and network channels for heavy ResNet (He et al., 2016). BATS (Bulat et al., 2020) devises a binary search space and incorporates it within the DARTS framework (Liu et al., 2018a). The aforementioned methods concentrate on the quantization of heavy networks, like ResNet (He et al., 2016), or replace the depthwise convolution with group convolution. Moreover, they inherit a two-stage search-retrain schema: once the best-quantized architectures have been identified, they need to be retrained for deployment. This procedure significantly increases the computational cost for the search if we have different deployment constraints and hardware bit-widths. Compared with all these methods, our OQA can search for quantized compact models and learn their quantized weights at the same time without additional retraining. Without our bit inheritance mechanism, these approaches also suffer from a significant drop of accuracy when a network is quantized to ultra-low bit-widths like 2.

## 3   METHOD

### 3.1   OVERVIEW

Our OQA aims to obtain compact quantized models that can be directly sampled from quantized supernet without retraining. As shown in Figure 1(c), the overall procedure of OQA is as follows: Step 1, Quantized Supernet Training (Section 3.3): Train a $K$-bit supernet by learning the weight parameters and quantization parameters jointly. Step 2: given a constraint on computational complexity, search for the architecture with the highest quantization performance on the validation dataset. If $K = N$, the whole process is finished. Step 3, Bit Inheritance (Section 3.4): Use the weight and quantization parameters of the $K$ bit supernet to initialize the weight and quantization parameters of the $K - 1$ bit supernet. Step 4: $K \leftarrow K - 1$ and Go to step 1.

The starting bit-width $K$ and the ending bit-width $N$ of the bit-inheritance procedure can be arbitrary. In this paper, we focus on quantized compact models under one fixed low bit-width quantization strategy, thus the starting bit-width and ending bit-width is 4 and 2.

### 3.2   PRELIMINARIES

**Neural Architecture Search without Retraining.**   Recently, several NAS methods (Yu et al., 2018; Cai et al., 2019; Yu et al., 2020) are proposed to directly deploy subnets from a well-trained supernet without retraining. Specifically, a supernet with the largest possible depth (number of blocks), width (number of channels), kernel size, and input resolution is trained. Then the subnet with top accuracy is selected as the searched network among the set of subnets satisfying a given computational complexity requirement. A subnet is obtained from parts of the supernet with depth, width, and kernel size smaller than the supernet. The subnet uses the well-trained parameters of the supernet for direct deployment without further retraining.

**Quantization Neural Network Learning.** To enable the training of quantized supernets, we choose a learnable quantization function following the recent state-of-the-art quantization method LSQ (Esser et al., 2019). In the forward pass, the quantization function turns the floating-point weights and activation into integers under the given bit-width. Given the bit-width $K$, the activation is quantized into unsigned integers in the range of $[0, 2^K - 1]$ and weights are quantized into signed integers in the range of $[-2^{K-1}, 2^{K-1} - 1]$. Given floating-point weights or activation $\mathbf{v}$, and learnable scale $s$, the quantization function $Q$ and its corresponding approximate gradient using the straight-through estimator (Bengio et al., 2013) is defined as follows:

$$\text{Quantization function: } \mathbf{v}^q = Q(\mathbf{v}, s) = \lfloor \text{clip}(\frac{\mathbf{v}}{|s|}, Q_{min}, Q_{max}) \rceil \times |s|,$$

$$\text{Approximate gradient: } \frac{\partial Q(\mathbf{v})}{\partial \mathbf{v}} \approx \mathbb{I}(\frac{\mathbf{v}}{|s|}, Q_{min}, Q_{max}), \tag{1}$$

where all operations for $\mathbf{v}$ are element-wise operations, $clip(z, r1, r2)$ returns $z$ with values below $r1$ set to $r1$ and values above $r2$ set to $r2$, $\lfloor z \rceil$ rounds $z$ to the nearest integer, $Q_{min}$ and $Q_{max}$ are, respectively minimum and maximum integers for the given bit-width $K$, $\mathbb{I}(\frac{\mathbf{v}}{|s|}, Q_{min}, Q_{max})$ means the gradient of $\mathbf{v}$ in the range of $(Q_{min} \times |s|, Q_{max} \times |s|)$ is approximated by 1, otherwise 0. $|s|$ returns the absolute value of $s$, which ensures that the semantics of scale $s$ is only interval, without inverting the signs of weights or activation. The scale $s$ is learned by back-propagation and initialized as $\frac{2}{\sqrt{Q_{max}}} \times |\bar{\mathbf{v}}|$, where $|\bar{\mathbf{v}}|$ denotes the mean of $|\mathbf{v}|$.

### 3.3 QUANTIZATION-AWARE NAS WITHOUT RETRAINING

**Existing problems of quantization-aware NAS.** Existing weight-sharing based quantization-aware NAS methods suffer from more unreliable order preserving (Guo et al., 2019; Bulat et al., 2020), as the quantization function introduces more instability on the learned weights. In our perspective, combining non-retrain NAS methods with quantization to avoid this problem can be a natural solution.

**Search space and quantized supernet.** Our search space is based on MobileNetV2 (Sandler et al., 2018) and MobileNetV3 (Howard et al., 2019), which has the flexible input resolution, filter kernel size, depth (number of blocks in each stage), and width (number of channels). Our search space consists of multiple stages. Each stage stacks several inverted residual blocks. Further details about search space can be found in Appendix A.5, A.6

Unlike the floating-point supernet training (Cai et al., 2019; Yu et al., 2020), we utilize the quantization function Eq. 1 to discretize the weights and activation values for the quantized supernet training. Meanwhile, the floating-point weights need to be retained for reducing quantization loss at the training stage of each bit-width. For a convolution layer with input activation $\mathbf{a}$ and weight $\mathbf{w}$, we define the corresponding learnable scales of activation and weight as $s_{\mathbf{a}}$ and $s_{\mathbf{w}}$. The forward pass for a quantized convolution layer is defined as follows:

$$\begin{aligned} \mathbf{w}^q &= Q(\mathbf{w}, s_{\mathbf{w}}), \\ \mathbf{a}^q &= Q(\mathbf{a}, s_{\mathbf{a}}), \\ \mathbf{y} &= \mathbf{w}^q * \mathbf{a}^q, \end{aligned} \tag{2}$$

where $Q(\cdot, \cdot)$ is the learnable quantization function defined in Eq. 1, * is the convolution operation and $\mathbf{y}$ is the output of this layer.

**Subnet sampling.** During the supernet training, different subnets are sampled and trained in each iteration. In non-retrain NAS methods, a subnet has a smaller scale than the supernet in resolution, width, depth, and kernel size, which can be obtained by cropping the corresponding part from the supernet. In our settings, a stage with $d$ blocks in a subnet inherits the weights from the first $d$ blocks in the same stage of the supernet. A depthwise convolution layer in a subnet with width $e$ and kernel

size $k$ are cropped from the central $k * k$ region of the first $e$ kernels in the supernet's corresponding convolution layer. The input image of each subnet is resized to its resolution $r$.

Our subnet sampling strategy combines the supernet training pipeline proposed in (Cai et al., 2019; Yu et al., 2020), and it has two stages. First, we only sample the biggest quantized subnet until it converges as in (Cai et al., 2019). Afterwards, we use the sandwich rules (Yu & Huang, 2019) to sample subnets which means in one iteration, the biggest subnet and the smallest subnet, and two random sampled subnets are sampled. Further details can be found in the Appendix A.2.

**Architecture search of quantized supernet.** We directly evaluate the sampled subnets from the supernet without further retraining. It's worth mentioning that we use the predictive accuracy on 10K validation images sampled from *trainset* to measure the subnets in the search procedure. Furthermore, we exploit a coarse-to-fine architecture selection procedure, similar to Yu et al. (2020). We first randomly sample 10K candidate architectures from the supernet with the FLOPs of the corresponding floating-point models ranging from 50M to 300M (2K in every 50M interval). After obtaining the good skeletons (input resolution, depth, width) in the pareto front of the first 10K models, we randomly perturb the kernel sizes to further search for better architectures.

### 3.4 Quantization-aware NAS with Bit Inheritance

**The problem of quantization-aware NAS with ultra-low bit-widths.** When the quantization bit is lower than 3 bit, the traditional quantization-aware training (QAT) (Kim et al., 2019; Bhalgat et al., 2020) process is highly unstable and introduces very large accuracy degradation for the challenging case of the 2 bit model. Using the approach introduced in Section 3.3, we obtain the quantized supernet with the highest $K = 4$ bit. To further obtain the quantized supernets of lower bit-widths (*e.g.* $K - 1$, $K - 2$), we can use the QAT to directly train quantized supernets for each bit-width. As shown in Table 1, the biggest architecture in our search space suffers a 19.1% accuracy drop between 4 bit and 2 bit when using the QAT. Besides, QAT requires much more computational cost.

**Inheritance from the high bit to low bit.** We propose a bit inheritance procedure to compensate for both the disadvantages. First, we use the $K$ bit supernet to provide a good initialization for the weights of $K - 1$ bit-width supernet. In the initialization for $K - 1$ bit supernet, we first inherit the weight parameters and scale parameters from $K$ bit supernet. Then the scale parameters are doubled because they map the floating-point values to the integer range of $K - 1$ bit, which is half the range of $K$ bit. Finally, to compensate for the statistics shift of each layer's output caused by quantization error, we forward the model to recalculate the mean and variance of the BatchNorm layers (Yu & Huang, 2019) with randomly sampled 4096 training images. During training, we use the $K$ and $K - 1$ bit supernets as teacher and student, and

Table 1: The accuracy of the biggest model in quantization-aware training(QAT) and progressive bit inheritance from high bit to low bit. Start denotes the accuracy with one epoch training, and end denotes the accuracy at the end of training.

|                      | 4/4   | 3/3   | 2/2   |
|----------------------|-------|-------|-------|
| QAT-Start            | 48.1% | 23.2% | 0.8%  |
| QAT-End              | 75.1% | 72.1% | 56%   |
| BitInheritance-Start | -     | 71.7% | 49.9% |
| BitInheritance-End   | -     | 72.7% | 63.9% |

train them in a knowledge distillation way (Hinton et al., 2015) to further reduce the quantization error between the $K - 1$ and $K$ bit parameters.

**The benefit of bit inheritance.** In the Appendix A.1, we show that the loss of weights of the $K - 1$ bit network is bounded (close to that of the $K$ bit network) if bit inheritance is used, where the parameters of the $K - 1$ bit network inherit from the parameters of the $K$ bit network. Therefore, bit inheritance help to guarantee the 2 bit network to be close to the 3 bit network in training loss. In comparison, existing methods like QAT start from a floating-point model which is far away from the 2 bit model and cause unstable training. The experimental results in Table 1 also validate the effectiveness of our design. The row named BitInheritance-Start means with only one epoch training, the initial accuracy is good enough in 3 bit. After finetuning with fewer epochs, the bit-inheritance achieves higher accuracy performance, especially for 2 bit.

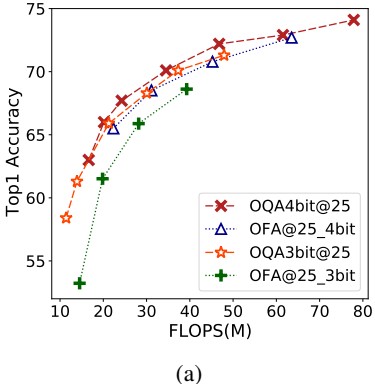 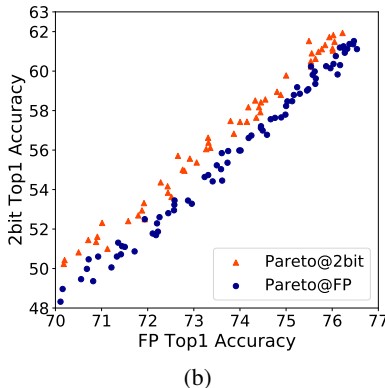

(a)          (b)

Figure 2: Comparison of the parato models of NAS-then-Quantize and OQA. (a) OFA FP supernet is used for NAS and LSQ is used as the quantization method. @25 denotes finetuning for 25 epoch. (b) The accuracy of Pareto@2bit/FP is obtained in the correponding 2bit/FP supernet.

# 4 EXPERIMENTAL ANALYSIS

## 4.1 EXPERIMENTAL SETTINGS AND IMPLEMENTATION DETAILS

We evaluate our method on the ImageNet dataset (Deng et al., 2009). If not specified, we follow the standard practice for quantized models (Kim et al., 2019; Gong et al., 2019) on quantizing the weights and activation for all convolution layers except the first convolution layer, last linear layer, and the convolution layers in the SE modules(Hu et al., 2018). To fairly compare with quantized compact models, the FLOPs are used and defined as follows. Denote the FLOPs of the FP layer by $a$, following (Zhou et al., 2016; Li et al., 2019b; Phan et al., 2020; Bulat et al., 2020), the FLOPs of the FP layer is $a$, and the FLOPs of $m$ bit weight and $n$ bit activation quantized layer is $\frac{mn}{64} \times a$, BitOPs is $mn \times a$ following (Wu et al., 2018). Unless otherwise noted, all results are sampled from MBV3 search space denoted as OQA, OQA-MBV2 represents the MBV2 search space. Further training details can be found in Appendix A.2.

## 4.2 NAS-THEN-QUANTIZE OR OQA

We denote pareto models as those models on the pareto front of the cost/accuracy trade-off curve. In Figure 2(a), we quantize the pareto models of the OFA floating-point supernet, which corresponds to the NAS-then-Quantize procedure in Figure 1(a). We compare it with the pareto models of our OQA that are directly obtained from the quantized supernet. In the comparison under 3 bit, the pareto curve of our OQA is far above that of the NAS-then-Quantize.

In Figure 2(b), we sample 10k subnets from the search space, and we validate these architectures from the FP supernet and 2 bit supernet. The pareto front of the subnets denoted as Parato@FP are selected with the FP accuracy and Pareto@2bit are selected with the 2 bit accuracy. With the same accuracy of the floating-point models, the accuracy of the model from the 2 bit pareto is higher than the model from the FP pareto. If our target is to search architectures for the quantized models, Figure 2(b) shows that searching from the quantized supernet as our OQA did is better than searching from FP supernet and then quantize. The advantage is more evident for lower bit-widths. Further details can be found in Appendix A.3.

## 4.3 EXISTING QUANTIZATION-AWARE NAS OR OQA

In Table 2, we compare our OQANet model family with several quantization-aware NAS methods, named SPOS (Guo et al., 2019), BMobi (Phan et al., 2020), BATS (Bulat et al., 2020) and APQ (Wang et al., 2020).

With the non-retraining quantized supernets, our OQANet model family shows great advantages over traditional weight-sharing methods corresponding to the paradigm of Figure 1(b). While SPOS (Guo et al., 2019) focuses on the search of network channels and bit-width of heavy

Table 2: Quantization-aware NAS performance under different bit-widths on ImageNet dataset. *Bit (W/A)* denotes the bit-width for both weights and activation. The number of bit for different layers is different for SPOS (Guo et al., 2019) with bit-width in the range of $\{1, 2, 3, 4\}$ and APQ (Wang et al., 2020) with bit-width in the range of $\{4, 6, 8\}$. BMobi (Phan et al., 2020), BATS (Bulat et al., 2020), and OQA use the same bit-width for different layers.

| Methods | Models | Bit (W / A) | FLOPs (M) | BitOPs (G) | Top-1 Acc.(%) |
|---------|--------|-------------|-----------|------------|---------------|
| SPOS | ResNet-34 | $\{1, 2, 3, 4\}$ | 337 | 13.11 | 71.5 |
| SPOS | ResNet-18 | $\{1, 2, 3, 4\}$ | 221 | 6.21 | 66.4 |
| BATS | 2× | 1 | 155 | 9.92 | 66.1 |
| BATS | 1× | 1 | 98.5 | 6.30 | 60.4 |
| BMobi | M2 | 1 | 62 | 3.97 | 59.3 |
| BMobi | M3 | 1 | 33 | 2.11 | 51.1 |
| **OQA** | OQA3bit-L | 3 | 48 | 3.07 | **71.3** |
| **OQA** | OQA3bit-M | 3 | 30 | 1.92 | **68.3** |
| **OQA** | OQA2bit-M | 2 | 19 | 1.21 | **61.7** |
| APQ | APQ-B | $\{4, 6, 8\}$ | 258 | 16.5 | 74.1 |
| APQ | APQ-A | $\{4, 6, 8\}$ | 206 | 13.2 | 72.1 |
| **OQA** | OQA4bit-L-MBV2 | 4 | 145 | 9.28 | **74.1** |
| **OQA** | OQA4bit-M-MBV2 | 4 | 107 | 6.85 | **72.4** |

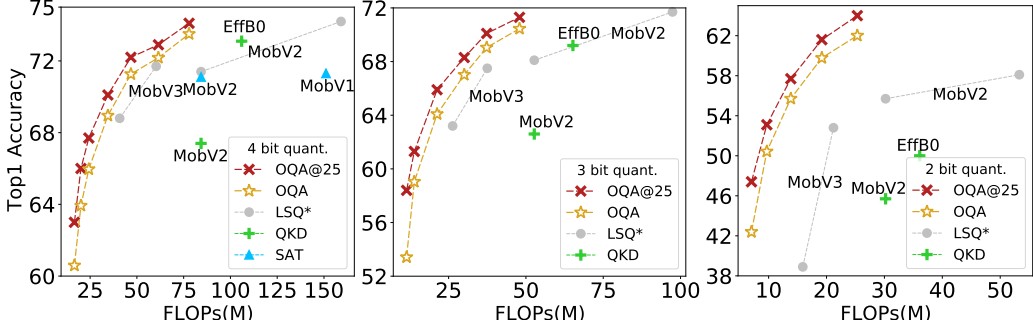

Figure 3: Comparison with the state-of-the-art quantization methods (LSQ , QKD, SAT) in various network(MobileNetV2/V3, EfficientNet-B0) on the ImageNet dataset.

ResNet (He et al., 2016), we focus on the search of compact models with fixed bit-width and achieve better results with fewer FLOPs. BMobi (Phan et al., 2020) and BATS (Bulat et al., 2020) did not provide the results for 2 bit, 3 bit or 4 bit. Therefore, we would like not to directly compare our approach with BMobi and BATS, because the results are obtained from different bit-widths. However, if only the FLOPs-accuracy trade-off is concerned, our OQA with a higher bit-width can be a better solution. APQ corresponds to the NAS-then-Quantize paradigm in Figure 1(a). It first searches for floating-point network architecture in the FP supernet, then trains a quantization-aware predictor to predict the searched quantized architecture. The transfer learning from FP predictor to quantized predictor brings the proxy problem and it also needs retraining. Our OQA4bit-L-MBV2 uses 43.7% less computation cost while maintaining the same accuracy as APQ-B.

### 4.4 FURTHER COMPARISON WITH NAS-THEN-QUANTIZE METHODS

We compare with several strong quantization methods including LSQ (Esser et al., 2019), LSQ+ (Bhalgat et al., 2020), APOT (Li et al., 2019b), QKD(Kim et al., 2019), SAT (Jin et al., 2019b), and LSQ* which is the LSQ implemented by us on different models to construct strong baselines. The result of 4 bit ResNet-18@LSQ* validates that our implementation is comparable.

Our OQA benefits from joint quantization and network architecture search, as well as the bit inheritance for lower bits. As shown in Table 3, our OQANets outperforms multiple quantization methods on models like MobileNetV2 (Sandler et al., 2018), EfficientNet-B0 (Tan & Le, 2019) and MbV3 (Howard et al., 2019) under all bit-widths we implements. **4 bit:** OQA4bit-L has 1% accuracy gain higher than Efficient-B0@QKD. OQA4bit-M outperforms ResNet-18@LSQ with 10%

Table 3: ImageNet performance under 4, 3, 2 bit. OQA4bit-M and OQA4bit-L denote medium and large model size in the 4 bit OQANets family respectively. @25 means we take weights from the supernet and finetune for 25 epochs. W/A denotes the bit-width for both weights and activation.

| Models | Method | Bit (W / A) | FLOPs (M) | Top-1 Acc.(%) |
|---|---|---|---|---|
| Efficient-B0 | QKD | 4 | 106 | 73.1 |
| **OQA4bit-L** | OQA@25 | 4 | 73 | **74.1** |
| ResNet-18 | LSQ / LSQ* | 4 | 542 / 542 | 71.1 / 70.9 |
| MobileNetV2 | LSQ* / SAT | 4 | 85 | 71.3 / 71.1 |
| MbV3-L (1.0x) | LSQ* | 4 | 60 | 71.7 |
| **OQA4bit-M** | OQA@25 | 4 | 47 | **72.3** |
| ResNet-18 | LSQ / APOT | 3 | 357 / 298 | 70.6 / 69.9 |
| Efficient-B0 | QKD | 3 | 65 | 69.2 |
| **OQA3bit-L** | OQA@25 | 3 | 48 | **71.3** |
| MobileNetV2 | LSQ* / QKD | 3 | 53 / 53 | 68.2 / 62.6 |
| MbV3-L 1.0x | LSQ* | 3 | 38 | 67.5 |
| **OQA3bit-M** | OQA@25 | 3 | 30 | **68.3** |
| Efficient-B0 | QKD / LSQ+ | 2 | 36 | 50.0 / 49.1 |
| MobileNetV2 | LSQ* / QKD | 2 | 30 | 55.7 / 45.7 |
| **OQA2bit-L** | OQA@25 | 2 | 25 | **64.0** |
| **OQA2bit-S** | OQA@25 | 2 | 14 | **57.7** |

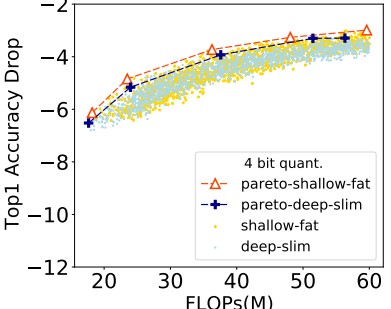 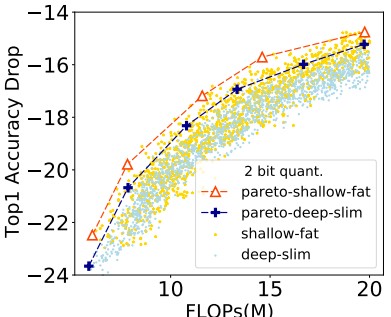

Figure 4: The quantization accuracy drop of shallow-fat and deep-slim subnets which are sampled from FP 4/2 bit supernets on the ImageNet dataset.

of its FLOPs. **3 bit:** Our OQA3bit-L can also match the accuracy of 3 bit ResNet-18@LSQ with 13% FLOPs and 3 bit Efficient-B0@QKD with 74% FLOPs. **2 bit:** Our OQA2bit-L requires less FLOPs but achieves significantly higher Top-1 accuracy (64.0%) when compared with EfficientNet-B0@QKD (50.0%) and MobileNetV2@LSQ* (55.7%). The results verify that the joint design of quantization and NAS results in more quantization-friendly compact models. We also show more searched models in the Figure 3 and our OQANets significantly outperform other quantization methods.

### 4.5    SHALLOW-FAT OR DEEP-SLIM

With the different quantized supernet obtained by the bit inheritance process, we analyze the following factors of a network: depth($D$), kernel size($K$), expand width($E$) and bits($B$). According the intuitive shape, we divide the models into two groups: 1) **shallow-fat:** $D$, $K$ and $E$ are sampled from $\{2,3\}$, $\{5,7\}$ and $\{4,6\}$ respectively, 2) **deep-slim:** $D$, $K$, and $E$ are sampled from $\{3,4\}$, $\{3,5\}$ and $\{3,4\}$ respectively.

Following the rules above, we randomly generate 1.5K architectures and obtain the accuracy of the floating-point and quantized models from the corresponding supernet. The input resolution is also randomly sampled. For a certain model, we use the accuracy drop from floating-point models to quantized models to measure the tolerence of quantization. As shown in Figure 4, the shallow-fat models are more quantization-friendly than deep-slim models. When the 4 bit and 2 bit situations are compared, the trend becomes more obvious when the quantization bits are lower.

Table 4: Top-1 Accuracy on ImageNet dataset under different settings. FLOPs is calculated with the corresponding floating-point models. Models named Nas-then-Quantize is chosen from the pareto front with the floating-point performance while the other is chosen with 2 bit performance. QAT@150 means the existing routine of quantization-aware training with 150 epochs.

| Models | FLOPs | Top-1 Acc.(%) FP Supernet | Top-1 Acc.(%) BI 2 bit Supernet | Top-1 Acc.(%) QAT@150 | Top-1 Acc.(%) QAT@500 |
|---|---|---|---|---|---|
| Nas-then-Quantize | 144 | 73.6% | 54.4% | 28.1% | 43.5% |
| **Joint Design** | 142 | 73.4% | **56.1**% | 33.2% | 47.2% |

Table 5: The comparison of the search cost and retrain cost with existing quantiztion-aware nas methods. $N$ denotes the number of models to be deployed. The total cost is calculated with $N = 40$.

| Methods | SPOS | BMobi | BATS | APQ | **OQA** |
|---|---|---|---|---|---|
| search cost (GPU hours) | $288 + 24N$ | $29N$ | $6N$ | $2400 + 0.5N$ | $\mathbf{1200 + 0.5N}$ |
| retrain cost (GPU hours) | $240N$ | $256N$ | $75N$ | $30N$ | **0** |
| total cost (GPU hours) | $10.8k$ | $11.4k$ | $3.2k$ | $3.6k$ | $\mathbf{1.2k}$ |

## 4.6 ABLATION STUDY

**The effectiveness of joint design.** In Table 4, the top-1 accuracy on ImageNet under different settings is listed. NAS-then-Quantize means the model is selected from the pareto front with the floating-point accuracy, and Joint Design is with the 2 bit accuracy. With similar FLOPs, similar floating-point accuracy, Joint Design surpass the accuracy of Nas-then-Quantize with $1.7\%$ in 2 bit accuracy. We also perform the quantization-aware training(QAT) with 150 epochs and the joint design results in over $5\%$ accuracy improvement, which verifies the effectiveness of Joint Design in finding quantization-friendly architectures.

**The effectiveness of bit inheritance.** In Table 4, we further compare these two models which performs quantization-aware training(QAT) with 150 epochs and 500 epochs. In QAT, pretrained quantized floating-point models are used to provide a better initialization point, while our bit inheritance proposes to initialize models with one bit-width higher. Although 2 bit models benefit from QAT with more epochs, the bit inheritance (BI) supernet still improves the two models' accuracy by a large margin, even compared to training with pretrained floating-point weights by 500 epochs. The huge accuracy improvement verifies that bit inheritance is a better practice compared with the existing routine of quantization-aware training because it alleviates the problem that quantized compact models with low bit-width are highly unstable to train.

**The efficiency of OQA framework.** In Table 5, we compare the search cost and the retrain cost of OQA with existing methods. The search cost is defined as the time cost of the supernet training and the search process to get the final $N$ searched models under latency targets. The retrain cost is defined as the training cost to get the final accuracy of the searched architecture. When $N$ is larger than 5, the retrain cost of SPOS (Guo et al., 2019) and BMobi (Phan et al., 2020) already surpasses the total cost of OQA. And We reduce at least half of the total cost compared with APQ (Wang et al., 2020). APQ needs to train a once-for-all floating-point supernet and sample thousands of FP subnets to perform quantization-aware training, and thus requires the transfer learning from the floating-point predictor to quantization predictor. Our OQA only needs to train one once-for-all quantized supernets and support a huge search space with over $10^{19}$ subnets that can be directly sampled from supernet without retraining. Thus, the average computational cost is relatively low.

## 5 CONCLUSION

In this paper, we present Once Quantized for All (OQA), a novel framework that deploys the searched quantized models without additional retraining and solves the problem of large accuracy degradation with bit inheritance mechanism under ultra-low bit-widths. With our proposed methods, we can search for the OQANet model family which far exceeds the existing quantization-aware NAS and quantization methods. Our results reveal the potential of quantized compact models under ultra-low bit-width.

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

# A  APPENDIX

## A.1  EFFECTIVENESS OF BIT INHERITANCE

Given a convolution layer in the $K$ bit supernet, we denote $s_K$ as the scale parameters of this layer, and $s_{K-1} = 2s_K$ as the doubled scale for the $K-1$ bit supernet. We use $\mathbf{w}$ and $N_{\mathbf{w}}$ to denote the weights of this layer which is inherited from $K$ to $K-1$. Next, we show that the $L_1$ distance of $Q(\mathbf{w}, s_K)$ and $Q(\mathbf{w}, s_{K-1})$ is bounded by $N_{\mathbf{w}} \cdot s_K$. It means the initialized $Q(\mathbf{w}, s_{K-1})$ has a bounded distance with the well-trained $Q(\mathbf{w}, s_K)$.

For each $w_i$, we have:

$$Q(\mathbf{w_i}, s_K) = \lfloor \text{clip}(\frac{\mathbf{w}_i}{|s_K|}, -2^{K-1}, 2^{K-1} - 1) \rceil \times |s_K|,$$

$$Q(\mathbf{w_i}, s_{K-1}) = \lfloor \text{clip}(\frac{\mathbf{w}_i}{|s_{K-1}|}, -2^{K-2}, 2^{K-2} - 1) \rceil \times |s_{K-1}| \tag{3}$$

$$= 2\lfloor \text{clip}(\frac{\mathbf{w}_i}{|2s_K|}, -2^{K-2}, 2^{K-2} - 1) \rceil \times |s_K|.$$

Based on this expression, we further get:

$$|Q(\mathbf{w_i}, s_K) - Q(\mathbf{w_i}, s_{K-1})| =$$
$$|\lfloor \text{clip}(\frac{\mathbf{w}_i}{|s_K|}, -2^{K-1}, 2^{K-1} - 1) \rceil - 2\lfloor \text{clip}(\frac{\mathbf{w}_i}{|2s_K|}, -2^{K-2}, 2^{K-2} - 1) \rceil| \times |s_K|. \tag{4}$$

For any $\mathbf{w_i}$ and $s_K$, we have:

$$|\lfloor \text{clip}(\frac{\mathbf{w}_i}{|s_K|}, -2^{K-1}, 2^{K-1} - 1) \rceil - 2\lfloor \text{clip}(\frac{\mathbf{w}_i}{|2s_K|}, -2^{K-2}, 2^{K-2} - 1) \rceil| \leq 1. \tag{5}$$

Thus,

$$|Q(\mathbf{w_i}, s_K) - Q(\mathbf{w_i}, s_{K-1})| \leq |s_K|,$$
$$\text{and} \tag{6}$$
$$||Q(\mathbf{w}, s_K) - Q(\mathbf{w}, s_{K-1})||_1 \leq N_w \cdot |s_K|.$$

## A.2  TRAINING DETAILS

**Dataset config:**  We evaluate our method on the ImageNet dataset. The training dataset is made up of 1.28 million images with resolution $224 \times 224$ belonging to 1000 classes and the validation set has 50k images. For ImageNet training, we use the typical random resized crop, randomly horizontal flipping and color jitter of $[32/255, 0, 0.5, 0]$ for data augmentation. During evaluation, we first determine the active image size $s$, and resize the image into $\lceil s/0.875 \rceil \times \lceil s/0.875 \rceil$ and center crop $s \times s$ image.

**Quantization aware training:**  We reimplement LSQ (Esser et al., 2019) as our base quantization method. We start from a floating-point model and finetune the model for 150 epochs. The optimizer is SGD with Nesterov momentum 0.9 and weight decay 3e-5, and the label smoothing ratio is 0.1. The initial learning rate is 0.04 under the batch of 1024, with the cosine annealing schedule. The dropout rate is 0.1. Except for learning rate and training epochs, we follow this protocol in the OQA procedure.

**OQA procedure:**  Combining the advantages of OFA (Cai et al., 2019) and BigNas (Yu et al., 2020), the overall OQA procedure is divided into four steps as follow:

Step0: we train the 4 bit biggest models in the search space. It follows the typical quantization-aware training, we use the floating-point pre-trained model as initialization and finetuning for 150 epochs. The learning rate is 0.04 with a batch-size of 1024.

Step1: in the supernet training phase, the biggest model obtained in step 0 is used as initialization. The input resolution, kernel size, width, and depth are randomly sampled. This whole process takes

200 epochs. In one iteration, four models are sampled with the sandwich rule Yu & Huang (2019), which is the biggest subnet and the smallest subnet, and two random sampled subnets. The learning rate is 0.02 with a batch size of 1024.

With bit inheritance, the training of the 3/2 bit supernet is simplified. And the training time is reduced.

Step2: In the 3 bit supernet, we use the 4 bit supernet obtained in Architecture shrinking Step1 part2 as initialization. We directly random sample input resolution, kernel, width, and depth. four models are sampled, which is the biggest subnet and the smallest subnet, and two random sampled subnets for one update. We only use 25 epochs and the learning rate is 0.0016.

Step3: In the 2 bit supernet, we use the 3 bit supernet obtained in Step2 as initialization. We also directly random sample resolution, kernel, width, and depth. Four models are sampled, which is the biggest subnet and the smallest subnet, and two random sampled subnets for one update. We only use 120 epochs and the learning rate is 0.0256.

**OQA subnet finetuning:** Our OQA performance can be further improved by finetuning the subnet weights sliced from the OQA supernet as suggested by OFA (Cai et al., 2019). The accuracy of the subnet is already higher than training from scratch. In default, the subnets are finetuned for 25epochs. The initial learning rate is 0.0016 with a batch size of 1024, with the cosine annealing schedule.

**Knowledge distillation:** The knowledge distillation(KD) used in our experiment is the traditional loss(KD) proposed in (Hinton et al., 2015). The student's logits and teacher's logits are used to calculating the cross-entropy loss. The temperate is 1 and the kd loss weight is 1.

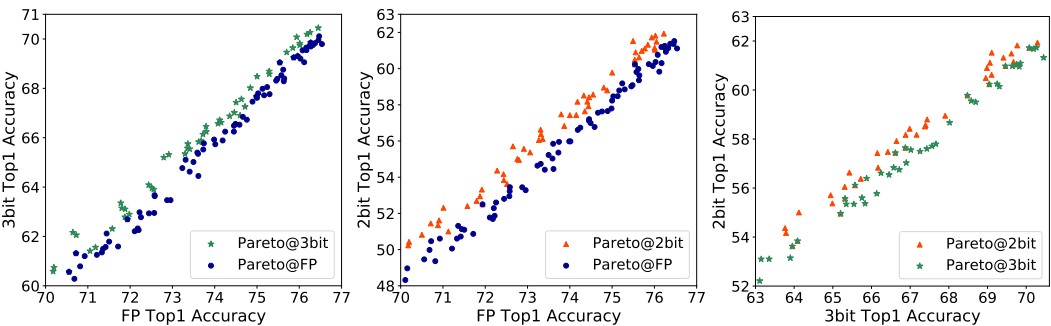

Figure 5: Comparison of the Top-1 validation accuracy on ImageNet dataset between the FP pareto, 3 bit pareto, and 2 bit pareto. The pareto is selected from the corresponding supernet and the accuracy is also obtained from the supernet.

### A.3 NAS-THEN-QUANTIZE OR OQA

In Figure 5, we sample 10k subnets from the search space, and we validate these subnets from the FP supernet, 3bit, and 2 bit supernet. The pareto front of the subnets denoted as Parato@FP is selected with the accuracy in the floating-point supernet and Pareto@3bit/Pareto@2bit is selected with the accuracy in the 3 bit/2 bit supernet. In Figure 5, the first two figure reveals that as the bit decreases, the accuracy gain increases with searching directly in the quantized supernet. We further compare the Pareto@3bit and Pareto@2bit, and it shows that with the same 3 bit accuracy, the accuracy of the model from the 2 bit pareto is higher than the model from the 3 bit pareto, but the accuracy gain is less compared with the FP pareto. To search for 2 bit quantized models, it is best to search directly in the 2 bit supernet, and it is better to search in the 3 bit supernet than searching in the FP supernet.

Observed the difference in the accuracy of the pareto architectures in different supernets at the same 2 bit, we are

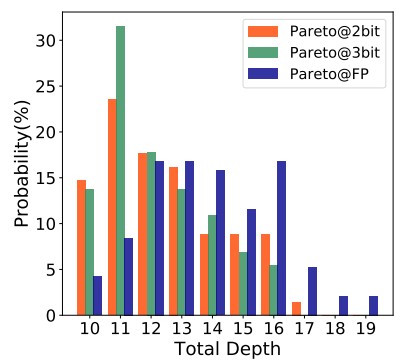

Figure 6: The probability distribution of total depth.

curious whether the accuracy performance is attributed to the structure difference in the pareto architecture. In Figure 6, we plot the distribution of the total depth of the pareto subnets from different supernet. In the depth dimension, the distribution reveals that Pareto@FP favors deeper models while Pareto@3bit/2bit favors shallow models. And the distribution of depth is closer between Pareto@3bit and Pareto@2bit.

## A.4 The details of quantization-aware NAS.

Table 6: The details of quantization-aware nas named SPOS (Guo et al., 2019), BMobi (Phan et al., 2020), BATS (Bulat et al., 2020), APQ (Wang et al., 2020) are given, including network architecture, search space, bit-width and quantization algorithm PACT (Choi et al., 2018), Bireal (Liu et al., 2018b), Xnor-net++ (Bulat & Tzimiropoulos, 2019), HAQ (Wang et al., 2019), LSQ (Esser et al., 2019). Group MobileNet denotes the MobileNet with group convolution in place of depthwise convoluton.

| | SPOS | BMobi | BATS | APQ | OQA |
|---|---|---|---|---|---|
| Quantization Algorithm | PACT | Bireal | Xnor-net++ | HAQ | LSQ |
| Network Architecture | ResNet | Group MobileNet | Group Darts | MobileNetV2 | MobileNetV2, MobileNetV3 |
| Bit Width | $\{1, 2, 3, 4\}$ | $\{1\}$ | $\{1\}$ | $\{4, 6, 8\}$ | $\{2, 3, 4\}$ |
| Search Space | width, bit-width | group number | operation, connection | width, depth, kernel size, bit-width | width, depth, kernel size, resolution |
| Retrain | ✓ | ✓ | ✓ | ✓ | ✗ |

## A.5 MobileNetV2 Search Space

Table 7: MBConv refers to inverted residual block which has a '$1 \times 1$ pointwise - $k \times k$ depthwise- $1 \times 1$ pointwise' structure without SE module (Hu et al., 2018), MBConv-SE is the MBConv block with SE module. Channels means the number of output channels in this stage. Depth means the number of blocks or layers in this stage. Expand ratio refers the expand ratio of input channels which controls the width of the depthwise convolution. Convolution layers in the first and last has no expand ratio. Kernel size refers to the kernel size $k$ of the depthwise convolution.

| Stage | Operator | Resolution | Channels | Depth | Expand ratio | Kernel size |
|---|---|---|---|---|---|---|
| | Conv | $128 \times 128$ - $224 \times 224$ | 32 | 1 | | 3 |
| 1 | MBConv | $64 \times 64$ - $112 \times 112$ | 16 | 1 | 1 | 3 |
| 2 | MBConv | $64 \times 64$ - $112 \times 112$ | 24 | 2, 3, 4 | 3, 4, 6 | 3, 5, 7 |
| 3 | MBConv | $32 \times 32$ - $56 \times 56$ | 40 | 2, 3, 4 | 3, 4, 6 | 3, 5, 7 |
| 4 | MBConv | $16 \times 16$ - $28 \times 28$ | 80 | 2, 3, 4 | 3, 4, 6 | 3, 5, 7 |
| 5 | MBConv | $8 \times 8$ - $14 \times 14$ | 96 | 2, 3, 4 | 3, 4, 6 | 3, 5, 7 |
| 6 | MBConv | $8 \times 8$ - $14 \times 14$ | 192 | 2, 3, 4 | 3, 4, 6 | 3, 5, 7 |
| 7 | MBConv | $4 \times 4$ - $7 \times 7$ | 320 | 1 | 3, 4, 6 | 3, 5, 7 |
| | Conv | $4 \times 4$ - $7 \times 7$ | 1280 | 1 | | 1 |

## A.6 MobileNetV3 Search Space

Table 8: MBConv refers to inverted residual block which has a '$1 \times 1$ pointwise - $k \times k$ depthwise-$1 \times 1$ pointwise' structure without SE module (Hu et al., 2018), MBConv-SE is the MBConv block with SE module. Channels means the number of output channels in this stage. Depth means the number of blocks in this stage. Expand ratio refers the expand ratio of input channels which controls the width of the depthwise convolution. Convolution layers in the first and last has no expand ratio. Kernel size refers to the kernel size $k$ of the depthwise convolution.

| Stage | Operator | Resolution | Channels | Depth | Expand ratio | Kernel size |
|---|---|---|---|---|---|---|
| | Conv | $128 \times 128$ - $224 \times 224$ | 16 | 1 | | 1 |
| 1 | MBConv | $64 \times 64$ - $112 \times 112$ | 16 | 1 | 1 | 3 |
| 2 | MBConv | $64 \times 64$ - $112 \times 112$ | 24 | 2, 3, 4 | 3, 4, 6 | 3, 5, 7 |
| 3 | MBConv-SE | $32 \times 32$ - $56 \times 56$ | 40 | 2, 3, 4 | 3, 4, 6 | 3, 5, 7 |
| 4 | MBConv | $16 \times 16$ - $28 \times 28$ | 80 | 2, 3, 4 | 3, 4, 6 | 3, 5, 7 |
| 5 | MBConv-SE | $8 \times 8$ - $14 \times 14$ | 112 | 2, 3, 4 | 3, 4, 6 | 3, 5, 7 |
| 6 | MBConv-SE | $8 \times 8$ - $14 \times 14$ | 160 | 2, 3, 4 | 3, 4, 6 | 3, 5, 7 |
| | Conv | $4 \times 4$ - $7 \times 7$ | 960 | 1 | | 1 |
| | Conv | $1 \times 1$ | 1280 | 1 | | 1 |

## A.7 Architecture visualization

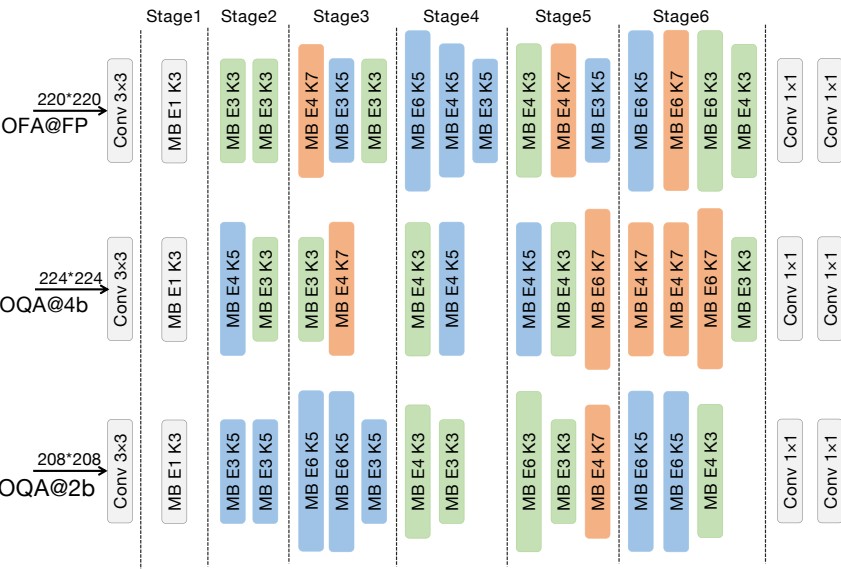

Figure 7: Architecture visualization of OFANet and our searched OQANets. 'MB E3 K3' indicates 'mobile block with expansion ratio 3, kernel size 3x3'. From top to bottom, there are FP OFANet, 4-bit OQANet and 2-bit OQANet. There are under similar computation cost, around 220M FP FLOPs.

