# OpenReview forum: "Once Quantized for All: Progressively Searching for Quantized Compact Models"
_ICLR.cc/2021/Conference — Reject_

### Official Review · AnonReviewer3 · 2020-10-26
**The paper shows promising results, but needs further polishing**

**Rating:** 4
**Confidence:** 4

**Review:**

Summary:
	This paper performs a joint optimisation for DNN models, making the NAS scheme is aware of both the quantisation and architectural search spaces. The paper presented a large range of comparisons to different quantisation strategies and ran a lot of experiments to support their claims. However, the writing quality of this paper is worrying. Also, I am a little worried about the novelty of this paper.

Strength:
1. There are a lot of experiments with the proposed method, showing a great empirical value for researchers in this field. I consider results shown in Figure 2 and Table 2 very supportive evidence of the effectiveness of the proposed method.
2. It is nice to see a large scale study (1.5K architectures) on some common properties of network architectures and their interactions with quantisation.
3. To my knowledge, this paper does present a state-of-the-art number for low-precision ImageNet classification.

Weakness:
1. The writing quality of this paper is worrying. This is not simply to do with the use of language, but also on the clarity of some matters. I strongly recommend the authors to have a serious polish of their paper, since they do present valuable results and STOA numbers.
2. To me, the novelty of this paper is limited, it seems like an extension to Once-for-all, and the authors also cited this work. The teacher-student technique is also a published idea. The authors claim this is the first piece of work of NAS without re-training. However, they are iteratively reducing the bit-width K, which implies a large training cost and is somehow equivalent to re-training. The method in the paper looks like a combination of a number of well-known techniques, which might limit the novelty claim in this paper. However, I have to say I am not very troubled with combining a bunch of existing techniques if it show new STOA that is outperforming by a significant margin. This weakness is only minor to me.

My suggestions & confusions:
1. It seems like you can boost the performance of quantised networks from a) jointly search for architectures and quantisation and b) teacher-student alike quantisation training with inherited weights. Could you test these two parts in isolation and quantify the contributions of each technique?
2. Why you quantise activations to unsigned numbers (Page 4)? Don’t you consider activations like leakyrelu in your activation search space? Or you do not search activations at all?
‘... NAS methods suffer from more unreliable order preserving’, Who are you comparing to in this case? Is it more unreliable compared to RL based NAS?
3. What is your Flops reported in Table 2? Flops means floating point operations, do you mean bitops? or you somehow scaled flops with respect to bitwidths?
4. ‘we focus on the efficient models under one fixed low bit-width quantization strategy’ Do you mean the network is uni-precision? So no layer-wise mixed-precision is allowed?
5. I spotted a number of misused languages, and will strongly recommend authors to check mistakes like:
a) Ambiguity:
 i) ‘with high floating-point performance’: do you mean floating-point models? Or you mean customised floating-point models? Describe floating-point as high is very misleading.
ii) ‘quantize the network with retraining’: I guess I understand what you mean, but you might say “retrain the quantised models” to be less ambiguous.

b) Grammar:
i). ‘different bit-width’ -> ‘different bit-widths’
ii) ‘quantization supernet’ -> ‘quantized supernet’ and so on.

c) Do not assume readers have prior knowledge:
i) ‘we use sandwich rules’ -> ‘we use the sandwich rule’ and maybe you should consider explain what it is.

I cannot present all the mistakes here, these are just examples, I would iterate again that I would strongly recommend you to polish the paper since I do like the results you are presenting and think if the code is open-sourced, they will benefit the community.

---

> ### Author Response · Authors · 2020-11-20
> **Response to AnonReviewer3**
>
> We sincerely thank you for the detailed feedback. We will seriously polish our paper and fix all the typos and the misused languages as pointed in the revised version. We will explain your concerns point by point.
>
> Q.1. To me, the novelty of this paper is limited, it seems like an extension to Once-for-all, and the authors also cited this work. The teacher-student technique is also a published idea. The authors claim this is the first piece of work of NAS without re-training. However, they are iteratively reducing the bit-width K, which implies a large training cost and is somehow equivalent to re-training. The method in the paper looks like a combination of a number of well-known techniques, which might limit the novelty claim in this paper. However, I have to say I am not very troubled with combining a bunch of existing techniques if it show new STOA that is outperforming by a significant margin. This weakness is only minor to me.
>
> A.1. We agree that our method is not making a major contribution to the literature of NAS. As we demonstrated in our paper, our OQA stands on the shoulder of a principle NAS without retraining method, e.g. OFA, BigNAS. Our efforts are mainly focused on how to combine the state-of-the-art NAS methods and quantization to see if we can benefit both sides. Surprisingly, we come up with the following conclusions which might give some insights to the quantization community: (1) the joint search with quantized once-for-all supernet results in quantized efficient models with high accuracy, especially, the accuracy gain increases as the bit decreases. (2) bit-inheritance alleviates the unstable training problem of quantized supernet under ultra-low bit-width, and improve the accuracy by a large margin. (3) with a huge search space under different bit-widths, we are able to evaluate the effect of the network factors which could provide some insight into quantization friendly architecture design. We further study the quantization friendliness of architectures with systematical analysis in Section 4.5, which reveals that shallow-fat models are indeed more likely to be quantization friendly than deep-slim models under low bit-width. Thus, we regard our work as practically useful and set up a nice baseline in the intersection area of quantization and network architecture search.
>
> As for the training cost with bit-inheritance to progressively reduce the bit-width, the bit-inheritance for 3/2bit supernets actually reduces the training cost compared with the 4bit quantized supernet training. The training cost of 3/2bit quantized supernet only needs 15% / 50% time cost of training 4bit quantized supernet if bit-inheritance is used. Bit-inheritance is aimed at pushing ahead the performance boundary of ultra-low bit quantized efficient models.
>
> Q.2. It seems like you can boost the performance of quantised networks from a) jointly search for architectures and quantisation and b) teacher-student alike quantisation training with inherited weights. Could you test these two parts in isolation and quantify the contributions of each technique?
>
> A.2. The results on `test these two parts in isolation and quantify the contributions of each technique' are shown in Table 1, Section 4.2, and Section 4.3 of the original paper. Specifically, in table 1, bit-inheritance results in 7.9% accuracy improvement under 2bit settings.
>
> Our OQA aims at pushing ahead the performance boundary of ultra-low bit quantized efficient models. Existing works seldom explore this challenging ultra-low bit setting. We conduct the ablation study about the benefit of joint design in Section 4.2 and bit inheritance in Section 3.4 individually in the rebuttal.
>
>
> | Models | FLOPs | FP Supernet | 2bit Supernet | QAT@150 | QAT@500 |
> | ---- | ---- |---- | ---- | ---- | ---- |
> | Nas-Then-Quantize  | 144 | 73.6% | 54.4% | 28.1% | 43.5% |
> | Joint Design | 142 | 73.4% | 56.1% | 33.2% | 47.2% |
>
> As demonstrated in the table above, joint design results in quantization-friendly architectures with over 5% accuracy gain in standalone quantization-aware training (QAT) with 150 epochs under the column of QAT@150. Although low bit-width models benefit greatly from more epochs training as shown in the column named QAT@500, the accuracy with the weights directly sampled from bit-inheritance supernet still surpasses QAT@500 by a large margin. The bit-inheritance alleviates the problem that low bit-width quantized efficient models are highly unstable to train.

---

> > ### Author Response · Authors · 2020-11-20
> > **Response to AnonReviewer3**
> >
> > Q.3. Why you quantise activations to unsigned numbers (Page 4)? Don’t you consider activations like leaky relu in your activation search space? Or you do not search activations at all?
> >
> > A.3. 1) In the original paper of LSQ, the activations are quantized to unsigned numbers. We genuinely followed their setting to ensure a fair comparison. 2) Thanks for bringing up the choice of activation function in our experiment. The concern of the fact that the choice activation function would affect the performance of quantized models were aware during our experiment. Since we would like to make our method easy to deploy on any given search space, we choose not to pick any activation function against the original search spaces(mbv2, mbv3, ...) used on NAS literature in the experiment. We believe this would give us a clean comparison between NAS method, Quantization method, and OQA.
> >
> >
> > Q.4. ‘... NAS methods suffer from more unreliable order preserving’, Who are you comparing to in this case? Is it more unreliable compared to RL based NAS?
> >
> > A.4. We refer to the existing quantization-aware NAS methods including RL based, weight-sharing based NAS methods. They all use proxies, e.g. smaller model or shared weights, to evaluate the subnets before retraining, while the ranking of the subnets tends to change after the retraining process. OQA avoids this by not requiring retraining.
> >
> > Q.5. What is your Flops reported in Table 2? Flops means floating point operations, do you mean bitops? or you somehow scaled flops with respect to bitwidths?
> >
> > A.5. We do not mean bitops. For Table 2, 1 floating point op is equal to one FLOPS. When this op is quantized with m-bit for weights and n-bit for activations, it is considered as m*n/64 FLOPS in table 2. For example, when m=n=3. Then the previous op is now 9/64 FLOPS. We follow [1][2][3] in this calculation. Bitops is 64 times of FLOPs in our definition. Details are provided in Section 4.1.
> >
> > Q.6. ‘we focus on the efficient models under one fixed low bit-width quantization strategy’ Do you mean the network is uni-precision? So no layer-wise mixed-precision is allowed?
> >
> > A.6. Yes. The network is uni-precision for our model in the experimental results. Our OQA framework is compatible with flexible bit-width training methods, such as Adabits. We use the same bit-width for all the layers because the layer-wise mix precision could be unfriendly to hardware optimization in real-time applications.
> >
> >
> > Q.7. I spotted a number of misused languages, and will strongly recommend authors to check mistakes like: a) Ambiguity: i) ‘with high floating-point performance’: do you mean floating-point models? Or you mean customised floating-point models? Describe floating-point as high is very misleading. ii) ‘quantize the network with retraining’: I guess I understand what you mean, but you might say “retrain the quantised models” to be less ambiguous. b) Grammar: i). ‘different bit-width’ -> ‘different bit-widths’ ii) ‘quantization supernet’ -> ‘quantized supernet’ and so on. c) Do not assume readers have prior knowledge: i) ‘we use sandwich rules’ -> ‘we use the sandwich rule’ and maybe you should consider explain what it is.
> >
> > A.7. We are very grateful for the detailed suggestions to help us polish our paper. All of these suggestions are valuable and right. And we fix as many typos as possible, refine misused language, and add the missing definition in th updated revision. The sandwich rule is explained in Appendix A.2 due to space issues.
> >
> > Q.8 If the code is open-sourced, they will benefit the community.
> >
> > A.8 Thanks for your appreciation.  We have uploaded the code and searched network architectures in the supplementary material.
> >
> > [1] Zhou S, Wu Y, Ni Z, et al. Dorefa-net: Training low bitwidth convolutional neural networks with low bitwidth gradients[J]. arXiv preprint arXiv:1606.06160, 2016.
> >
> > [2] Li Y, Dong X, Wang W. Additive powers-of-two quantization: A non-uniform discretization for neural networks[J]. arXiv preprint arXiv:1909.13144, 2019.
> >
> > [3] Bulat A, Martinez B, Tzimiropoulos G. BATS: Binary ArchitecTure Search[J]. arXiv preprint arXiv:2003.01711, 2020.

---

### Official Review · AnonReviewer4 · 2020-10-28
**Interesting paper, but some clarifications needed**

**Rating:** 6
**Confidence:** 2

**Review:**

This paper presents a new method to search for quantized neural networks. This method is different from others that it results in quantized weights which can be deployed without post-process such as fine-tuning. Proposed method first trains a 4-bit quantized supernet, and search for the best performance sub-net using the validation dataset. Then, the method initialize the 3-bit supernet using the 4-bit supernet, and trains 3-bit supernet using the knowledge distilation method. Proposed method iterates the initialization, training, and search process until the goal bit resolution is achieved.

I find that the idea of constructing quantized super-nets using Bit Inheritance is interesting and the paper is well written. However, I think more precise description of the training method and additional analysis is required to improve the paper.

1. In section 3.4, you described the K to K-1 supernet inheritance process as "During training, we use the K and K − 1 bit-width supernets as teacher and student, and train them in a knowledge distillation way to further reduce the quantization error between the K −1 and K bit-width parameters.". However, the knowledge distillation method is not specified after the statement. Is it similar to QKD? Or is it more of a traditional knowledge distillation approach? More precise description will be helpful.

2. In the introduction, you described that "... This two-stage procedure will undesirably increase the number of models to be retrained if we have multiple deployment constraints and hardware bit-widths...". However, there is no analysis on the search cost under such scenarios. Since the proposed method induces more supernet training process compared to NAS-then-Quantize approaches such as APQ, it is unclear whether your method will acheive lower search cost or not. I believe that additional analysis on the benefit of deploying the quantized weights without retraining in the mean of search cost must be given in the paper as one of the main contribution of the paper is that the proposed method allows the deployment without retraining.

3. Minor: In section 3.4, you described that "... where the parameters of the K − 1 bit network inherit from the parameters of the K − 1 bit network.". I think the later K - 1 must be K.

---

> ### Author Response · Authors · 2020-11-20
> **Response to AnonReviewer4**
>
> We sincerely thank you for your comprehensive comments and constructive advice. We will explain your concerns point by point.
>
> Q.1. However, the knowledge distillation method is not specified after the statement. Is it similar to QKD? Or is it more of a traditional knowledge distillation approach? More precise description will be helpful.
>
> A.1. We use the traditional cross-entropy loss[1] as our KD method, which is simpler than QKD. Specifically, the loss is calculated between the student’s logits and the teacher’s logits. Our work is orthogonal to other quantization-aware KD approaches, like QKD. We add a precise description in the uploaded revision in Appendix A.2.
>
> Q.2. There is no analysis of the search cost under such scenarios. Since the proposed method induces more supernet training process compared to NAS-then-Quantize approaches such as APQ, it is unclear whether your method will acheive lower search cost or not. I believe that additional analysis on the benefit of deploying the quantized weights without retraining in the mean of search cost must be given in the paper as one of the main contributions of the paper is that the proposed method allows the deployment without retraining.
>
> A.2. We have added the cost comparison in the uploaded revision in Section 4.6. According to the revised version in Table 2, the search cost and retrain cost with N models to be deployed are listed as follows:
>
> | Models | SPOS | BMobi | BATS | APQ | OQA |
> | ---- | ---- |---- | ---- | ---- | ---- |
> | search cost (GPU hours)  | 288+24N | 29N | 6N | 2400+0.5N | 1200+0.5N|
> | retrain cost (GPU hours) | 240N | 256N | 75N | 30N | 0|
> | total cost N=40 (GPU hours) | 10.8k | 11.4k | 3.2k | 3.6k | 1.2k|
>
> For our OQA, the search cost and retrain cost almost stay constant as the number of deployment scenarios N grows, while the cost of other approaches grows rapidly. In particular, with N=40, the total cost of OQA is 10x fewer than SPOS, BMobi, and 3x fewer than BATS and APQ. OQA supports a huge search space with billions of subnets that can be directly sampled from supernet without retraining. Thus, the average computational cost is relatively low.
>
>
> A.3 Minor: In section 3.4, you described that "... where the parameters of the K - 1 bit network inherit from the parameters of the K - 1 bit network.". I think the later K - 1 must be K.
>
> Q.3 Thank you for your precious advice. We will thoroughly proofread the paper. We have tried our best to fix the typos.
>
> [1] Hinton G, Vinyals O, Dean J. Distilling the knowledge in a neural network[J]. arXiv preprint arXiv:1503.02531, 2015.

---

### Official Review · AnonReviewer2 · 2020-10-28
**borderline**

**Rating:** 5
**Confidence:** 4

**Review:**

This paper proposed a method to train quantized supernets which can be directly deployed without retraining. The motivation is to have a supernet with a given quantization bit-width which only train once and can be deployed with different architectures (under different FLOPs budget). This paper made a bunch of experiments showing that the proposed once quantized for all method can find DNN architectures which have SOTA performance with low bit-width. The paper also shows that when training lower-bits supernet, it is helpful to use the weights from the trained higher-bits supernet.

Pros:

1. This paper targets a very practical problem that quantization is actually required for most resource-constrained devices. Recently, supernets (e.g. OFA, BigNAS) validate that it is possible to directly obtain DNNs with different FLOPs budget from a single big supernet without retraining. This saves lots of training time in the cases that one want to have DNNs with different FLOPs. Combining quantization-aware training and supernet is an effective approach to save training time when we want search different DNNs (with different size/FLOPs) which are quantized with certain bit-width.

2. The authors also show that it is very important to use the pretrained weights (from a trained higher-bits supernet) as initialization when training the quantized supernet. This observation is also meaningful for the cases that low-bits quantization-aware training is hard or unstable.

3. Using the proposed once-quantized-for-all method, the authors get several quantized DNNs which have SOTA results on ImageNet. The authors compared both "SOTA architectures + quantization aware training" and recent "quantization-aware NAS method". The accuracy / flops of the searched DNNs is better than the compared methods.

Cons:

1. It's natural to apply quantization-aware training on supernets when we want a supernet to be quantization-aware. Both quantization-aware training and supernets are ready-to-use techniques and the combination is straightforward. The benefit of bit-inheritance is a good observation, while using pretrained weights as initialization is kind of a common practice in quantization or model compression. At this point, the contribution in terms of the novelty is limited.

2. The proposed method can also outperform methods that can also search layer-wise bit-width (e.g., Table 2), although the method in this paper only uses the same bitwidth for all the layers. It's not clear what is the main factor in this comparison. Are all the methods using the same experiment setup? E.g., quantization algorithm (LSQ or min-max), architecture search spaces. It will be better to have an ablation study to understand which part of the proposed algorithm plays the key role to the better performance.


In general, I think this paper did a great job on the experiments of quantization-aware supernet, but the novelty contribution is slightly under the criteria of ICLR. So my rating is borderline. I hope the authors can give some response to the cons listed above, and I'd like to consider changing my rating if I missed something important.

---

> ### Author Response · Authors · 2020-11-20
> **Response to AnonReviewer2**
>
> We sincerely thank you for the valuable comments on our paper. We will explain your concerns point by point.
>
> Q.1. The benefit of bit-inheritance is a good observation, while using pretrained weights as initialization is kind of a common practice in quantization or model compression.
>
> A.1. We agree that using pre-trained weights as initialization is a common practice. However, direct usage of this common practice would be initializing K-1 bit-width supernet from 32 bit-width supernet, which does not work well if K-1 is small, e.g. 3 or 2. Our approach is different. The K-1 bit-width supernet is initialized from K bit-width supernet in bit-inheritance.
> And this is grounded with theory: Appendix A.1 proves that the weights inherited from K bit to K-1 have a bounded distance with the K bit weights. Therefore, the K-1 bit supernet is provided by an initialization point is near the well-trained K bit supernet, which cannot be achieved by initialization from 32 bit-with supernet. As another benefit, bit-inheritance can significantly reduce the cost of training new supernets.
>
>
> Q.2. The proposed method can also outperform methods that can also search layer-wise bit-width (e.g., Table 2), although the method in this paper only uses the same bit-width for all the layers.
>
> A.2. To the best of our knowledge, there are three reasons behind the phenomenon that OQA achieves more efficiency than the layer-wise bit-width search in SPOS with comparable performance: (1) SPOS suffers from the performance inconsistency between the evaluation with the shared weights and the retraining, while OQA does not require retraining; (2) OQA improves the performance further with bit inheritance; (3) The layer-wise bit-width search itself has not been proved to achieve a strong improvement, as mentioned in SPOS, only 1% accuracy gain is obtained with mixed-precision. Our OQA framework is compatible with flexible bit-width training methods, such as Adabits. We use the same bit-width for all the layers because the layer-wise mix precision could be unfriendly to hardware optimization in real-time applications. Moreover, OQA outperforms the layer-wise search method APQ in the same space, which can further verify the effectiveness of our proposed framework.
>
>
> Q.3. It's not clear what is the main factor in this comparison. Are all the methods using the same experiment setup? E.g., quantization algorithm (LSQ or min-max), architecture search spaces. It will be better to have an ablation study to understand which part of the proposed algorithm plays the key role to the better performance.
>
> A.3. Our OQA aims at pushing ahead the performance boundary of ultra-low bit quantized efficient models. Existing works seldom explore this challenging ultra-low bit setting. We have added the details of related methods with clarifications of quantization algorithm, network architecture, search space, and bit-width settings in the updated revision in Appendix A.4.
>
> |  | SPOS | BMobi | BATS | APQ | OQA |
> | ---- | ---- |---- | ---- | ---- | ---- |
> | quantization | PACT[1] | Bireal[2] | Xnor-net++[3] | HAQ[4] | LSQ[5] |
> | network | ResNet | Group MobileNet  | Group Darts | MobileNetV2 | MobileNetV2/3 |
> |search space | width | group number | operation/connection | width/depth/kernel size/bit-width |  width/depth/kernel size/resolution |
> | bit width | {1, 2, 3, 4} | {1} | {1} | {4, 6, 8} | {2, 3, 4} |
> | retrain | Yes | Yes | Yes | Yes | No |
>
> We also conduct the ablation study in Section 4.6 about the benefit of joint design in Section 4.2 and bit inheritance in Section 3.4 individually in the rebuttal.
>
> | Models | FLOPs | FP Supernet | 2bit BI Supernet | QAT@150 | QAT@500 |
> | ---- | ---- |---- | ---- | ---- | ---- |
> | Nas-Then-Quantize  | 144 | 73.6% | 54.4% | 28.1% | 43.5% |
> | Joint Design | 142 | 73.4% | 56.1% | 33.2% | 47.2% |
>
> As demonstrated in the table above, joint design results in quantization-friendly architectures with over 5% accuracy gain in standalone quantization-aware training(QAT) with 150 epochs under the column of QAT@150. Although low bit-width models benefit greatly from more epochs training as shown in the column named QAT@500, the accuracy with the weights directly sampled from bit-inheritance supernet still surpasses QAT@500 by a large margin with 8.9% accuracy gain.

---

### Official Review · AnonReviewer1 · 2020-10-30
**Jointly Network search and quantization outperforming state of the art results**

**Rating:** 6
**Confidence:** 3

**Review:**

This paper presents One Quantize for all framework. The framework claims to search for the network and the quantization without the need for retraining.

Results are promising although it is not clear to me if the comparisons are fair (different bit size).
Would also be great to see the resulting architecture and how it is different from other NAS approaches. It is not clear to me why the number of FLOPS is so low compared to related methods.

In the experiments, I missed the computational cost for training and how it compares to the other approaches (as this paper suggests there is no need for retraining or fine-tuning).

---

> ### Author Response · Authors · 2020-11-20
> **Response to AnonReviewer1**
>
> Thank you for your appreciation of our paper. We are glad to answer your doubts point by point.
>
>
> Q.1. It is not clear to me if the comparisons are fair (different bit size).
>
> A.1. Our OQA aims at pushing ahead the performance boundary of low bit-width quantized models. To verify the effectiveness of OQA, we compare it with a bundle of models, including manually designed models and NAS-powered models.
>
> Our setting is fair when considering the following two factors. 1) using the same bit-width in Table 3 when comparing with MbV3-1.0@LSQ and OQA4bit-M which has the same quantization algorithm, similar network architecture.  2) using the best practice for our approach. The table below shows that all existing works use different search spaces of architectures or bit-widths, quantization algorithms, weight initialization methods, and requirements of retrain. And it is difficult to use a single setting to compare with them because 1) code, time; 2) may not be fair since some designs are more suitable for some settings. We instead show that OQA is a good combination of the mentioned factors and verify the effectiveness of our quantized supernets and bit inheritance in the ablation study of our paper.
>
> |                         | SPOS     | BMobi | BATS |   APQ |  OQA |
> |  ----                  |   ----          |  ----         |  ----    |  ----    | ----    |
> | quantization  | PACT[1] | Bireal[2] | Xnor-net++[3] | HAQ[4] | LSQ[5] |
> | network        | ResNet   | Group MobileNet  | Group Darts | MobileNetV2 | MobileNetV2/3 |
> |search space | width | group number | operation/connection | width/depth/kernel size/bit-width |  width/depth/kernel size/resolution |
> | bit width     | {1, 2, 3, 4} | {1} | {1} | {4, 6, 8} | {2, 3, 4} |
> | retrain        | Yes | Yes | Yes | Yes | No |
>
>
> Q.2. Would also be great to see the resulting architecture and how it is different from other NAS approaches.
>
> A.2. Please refer to README.md in the supplementary material for the resulting architectures. We also visualize some searched architecture in Appendix A.7. OQANets tend to be shallower (fewer layers) and wider (more channels), while floating-point MobileNetV3 and OFANets are more likely to be deep and slim. We further study the quantization friendliness of architectures with systematical analysis in Section 4.5, which reveals that shallow-fat models are indeed more likely to be quantization friendly than deep-slim models under low bit-width. We assume that the quantization errors and gradient mismatch problem accumulate with more layers brought up with the quantization function.
>
>
> Q.3. It is not clear to me why the number of FLOPS is so low compared to related methods.
>
> A.3. Please note that, for the same kernel size and the same number of input and output channels, lower bit-width leads to lower FLOPS. For example, for the same architecture, the FLOPs are 28/45/162M under the bit-width of 3/4/32 respectively.
> Compared with the networks using the same bit-widths, e.g. in Table 2, OQANets have the same accuracy but lower FLOPs for two reasons: (1) While existing works aim at quantization with relative high bit-width(e.g. 4, 6, 8), OQA, powered by bit inheritance, is better at quantizing model with ultra-low bit-width(e.g. 2, 3, 4); (2) OQA obtains efficient and quantization friendly architectures with superior performance.
>
>
> Q.4. In the experiments, I missed the computational cost for training and how it compares to the other approaches (as this paper suggests there is no need for retraining or fine-tuning).
>
> A.4. We have added the cost comparison in the updated revision in Section 4.6. According to the revised paper in Table 2, the search cost and retrain cost with $N$ models to be deployed are listed as follows:
>
> | Models | SPOS | BMobi | BATS | APQ | OQA |
> | ---- | ---- | ---- | ---- | ---- | ---- |
> | search cost (GPU hours)  | 288+24N | 29N | 6N | 2400+0.5N | 1200+0.5N|
> | retrain cost (GPU hours) | 240N | 256N | 75N | 30N | 0|
> | total cost N=40 (GPU hours) | 10.8k | 11.4k | 3.2k | 3.6k | 1.2k|
>
> For our OQA, the search cost and retrain cost almost stay constant as the number of deployment scenarios $N$ grows, while the cost of other approaches grows rapidly. In particular, with $N$=40, the total cost of OQA is 10x fewer than SPOS, BMobi, and 3x fewer than BATS and APQ. OQA supports a huge search space with billions of subnets that can be directly sampled from supernet without retraining. Thus, the average computational cost is relatively low.

---

> > ### Author Response · Authors · 2020-11-20
> > **Response to AnonReviewer1**
> >
> > [1] Choi J, Wang Z, Venkataramani S, et al. Pact: Parameterized clipping activation for quantized neural networks[J]. arXiv preprint arXiv:1805.06085, 2018.
> >
> > [2] Liu Z, Wu B, Luo W, et al. Bi-real net: Enhancing the performance of 1-bit cnns with improved representational capability and advanced training algorithm[C]//Proceedings of the European conference on computer vision (ECCV). 2018: 722-737.
> >
> > [3] Bulat A, Tzimiropoulos G. XNOR-Net++: Improved binary neural networks[J]. arXiv preprint arXiv:1909.13863, 2019.
> >
> > [4] Wang K, Liu Z, Lin Y, et al. Haq: Hardware-aware automated quantization with mixed precision[C]//Proceedings of the IEEE conference on computer vision and pattern recognition. 2019: 8612-8620.
> >
> > [5] Esser S K, McKinstry J L, Bablani D, et al. Learned step size quantization[J]. arXiv preprint arXiv:1902.08153, 2019.

---

### Author Response · Authors · 2020-11-20
**Summary of Revision**

Dear reviewers, thank you for the detailed feedback and constructive advice! Some experiment results and changes have been added in the upload revision. We hope we have addressed all of your concerns. To clarify what has been changed, we have made the following overview of major changes.

1. We conduct the ablation study to further evaluate the effectiveness of our framework, the benefit of joint design, and the bit inheritance to progressively reduce the bit width.
2. The efficiency of our OQA framework. We compare the search and retrain cost with related quantization-aware NAS methods.
3. The details of knowledge distillation are illustrated in Appendix A.2.
4. We present the details of the related works with clarification of quantization algorithm, network architecture, search space, bit-width settings, and retrain or not in Appendix A.4.
5. We visualize three searched architecture under different bit-width in Appendix A.7.
6. We fix as many typos as we can, the misused language, and some missing definitions.

---

### Author Response · Authors · 2020-11-25
**Summary of Revision #2**

Dear reviewers, thank you again for the detailed feedback and we really appreciate your comments on this paper! We hope we have addressed all of your concerns in the rebuttal. And we try our best to polish our paper with the help of different advice. We sincerely hope that we do not make any misleading arguments. The overview of major changes is as follows.

1. we add BitOPs in table 2 in the main paper. And the definition is clarified in Section 4.1.
2. The sandwich rule is explained in paragraph "subnet sampling" of Section 3.3.
3. The definition of some shorthands is more specific in the caption such as the "Start and "End" in table 1.
4. We unify all "quantization training" to "quantization-aware training", or "quantization NAS" to "quantization-aware NAS", "quantization supernet" to "quantized supernet", and so on.
5. We refactor some expressions, for example, we actually commit large changes to the first three paragraphs in the introduction to make it clear.
6. In Section 3.1 overview, we add the description of ending bit-width N.
7. We correct all the misused language or missing definitions suggested by reviewers.
8. Fix some typos.

---

### Decision · Program_Chairs · 2021-01-07
**Final Decision**

**Decision:**

Reject

**Comment:**

This paper proposed a method to train quantized supernets which can be directly deployed without retraining. A main concern is that there is limited novelty. The proposed method looks like a combination of well-known techniques. Experimental results are promising. However, it is not clear if the comparisons are fair and if all the methods are using the same setup. It is desirable to have additional analysis and ablation studies. The writing can also be improved.